# Elucidate microbial characteristics in a full-scale treatment plant for offshore oil produced wastewater

**Shuyuan Deng[1], Bo Wang[1], Wenda Zhang[2], Sanbao Su[2], Hao Dong[2], Ibrahim M. Banat[3], Shanshan Sun[2], Jianping Guo[1], Weiming Liu[4], Linhai Wang[5], Yuehui She[2], Fan Zhang⬤[1]***

**1** School of Energy Resources, China University of Geosciences (Beijing), Beijing, China, **2** College of Petroleum Engineering, Yangtze University, Wuhan, Hubei, China, **3** Faculty of Life and Health Sciences, University of Ulster, Coleraine, N. Ireland, United Kingdom, **4** Sinopec Shengli Oilfield, Dongying, Shangdong, China, **5** CNOOC Energy Development Co. Ltd. Technology Branch, Beijing, China

* fanzhang@cugb.edu.cn

## Abstract

Oil-produced wastewater treatment plants, especially those involving biological treatment processes, harbor rich and diverse microbes. However, knowledge of microbial ecology and microbial interactions determining the efficiency of plants for oil-produced wastewater is limited. Here, we performed 16S rDNA amplicon sequencing to elucidate the microbial composition and potential microbial functions in a full-scale well-worked offshore oil-produced wastewater treatment plant. Results showed that microbes that inhabited the plant were diverse and originated from oil and marine associated environments. The upstream physical and chemical treatments resulted in low microbial diversity. Organic pollutants were digested in the anaerobic baffled reactor (ABR) dominantly through fermentation combined with sulfur compounds respiration. Three aerobic parallel reactors (APRs) harbored different microbial groups that performed similar potential functions, such as hydrocarbon degradation, acidogenesis, photosynthetic assimilation, and nitrogen removal. Microbial characteristics were important to the performance of oil-produced wastewater treatment plants with biological processes.

**Data Availability Statement:** The samples were collected by the State Oceanic Administration CNOOC Energy Development Co. Ltd (No. GC2018zZCYH0107). the RNA sequencing data

## Introduction

Oil and gas production is one of the most important industrial activities as the significance of oil and natural gas remains important to modern civilization. In general, after extraction from subsurface oil reservoirs, extracted oil-water liquids travel from wellbores through pipelines to three-phase separators where they are separated into gas, oil, and water phases. The water phase constitutes the major wastewater source of the oil industry. Owing to increasing environmental concerns, researchers have paid considerable attention to the effects dissolved oil components and toxic chemicals in oil-produced wastewater on living organisms and the environment [1–3]. Many countries have implemented stringent regulatory standards for discharging oilfield wastewater into natural environments. In China, the average limits of oil and

has been uploaded to NCBI Sequence Read Archive, and please use accession numbers SRR5482389 through SRR5482359 to access these data.

**Funding:** This study was supported by the National Natural Science Foundation of China in the form of grants awarded to FZ (51774257, 51504221) and YS (51574038, 51634008), the National Science and Technology Major Oil and Gas Special Project in the form of a grant awarded to YS (2017ZX05009-004), the Weizhou Terminal COD Treatment System Material Procurement Project in the form of a grant used to obtain the study samples (GC2018ZCYH0107), Sinopec Shengli Oilfield in the form of a salary for WL, and CNOOC Energy Development Co. Ltd. in the form of a salary for LW. The specific roles of these authors are articulated in the 'author contributions' section. The funders had no role in study design, data collection and analysis, decision to publish, or preparation of the manuscript.

grease discharge and chemical oxygen demand (COD) are 10 and 100 mg/L, respectively. Specifically, the EU Water Framework Directive, the Oslo Paris Convention, and oil operators in Norway are committed to the "zero discharge" of pollutants into the sea [4].

To meet the stringent standards of water discharge, many physical, chemical, and biological processes have been developed for the treatment of oil-produced wastewater [4]. Given its efficiency and cost-effectiveness, optimizing such treatment systems with combined technologies is necessary [5, 6]. Biological processes involving physical and chemical pretreatments are becoming increasingly popular. The efficiency of biological treatment processes largely depends on the activities of existing functional microbial communities [5]. Although some studies reported the performance of oil-produced wastewater treatments using biological processes, few focused on revealing microbial ecology because they were carried out in laboratory-scale bioreactors where bacterial consortia were mixtures of cultured and isolated microbes and produced wastewater was synthetic. These laboratory-scale studies without considering species-species and environments-species interactions that govern bacterial community development and dynamics. Even though, there were studies using natural produced water from oil fields [7–9] and reports monitoring field pilot produced water treatments in oil production plants [10, 11], few efforts are dedicated to elucidating the characteristics of developed and activated microbes. Thus, the biological fundamental behind the treatment technologies for oil produced wastewater are always in "black boxes" as we are just beginning to understand the diversity and biogeography of microbial communities in wastewater treatment plants [12].

In addition, the sequencing of 16S ribosomal (RNA) gene can profile intriguing microbial taxonomic community but often cannot explain microbial function variation. Recently, a database named Functional Annotation of Prokaryotic Taxa (FAPROTAX) has been constructed to map prokaryotic tax to metabolic or other ecologically relevant functions [13]. FAPROTAX has been evaluated thoroughly through direct comparison with metagenomics [13] and has been used successfully in elucidating the functional diversity in mangrove-inhabited mudflats and bromeliad tank microbiome [14]. However, despite its apparent merits, FAPPOTAX has limitations. A majority of detected sequences cannot be affiliated with functional groups. The reason is that many functions are only conserved at the species or genus level in the FAPROTAX database [13, 14], while many sequences detected by high-throughput sequencing methods are identified with bacteria up the order level. To narrow the data gap disregarded by FAPROTAX, we blasted the sequences of dominant OTUs with the NCBI database and extracted relevant information, such as environmental conditions, enrichment purpose, and potential functions, from sequences with the highest similarity found in the literature to assume the functions of the unidentified bacteria, which is the basic construction method for FAPROTAX database [13].

In this work, we elucidated the structures and potential functions of bacteria in a full-scale offshore oil produced wastewater treatment plant on the basis of high-throughput 16S rDNA sequences. Samples from treated streamline processes, including physical, chemical, and biological processes, were collected. The obtained 16S rDNA high-throughput sequences were blasted with the NCBI database (https://blast.ncbi.nlm.nih.gov/Blast.cgi) and were annotated to potential functions with the FAPROTAX software. Finally, we inferred the microbial characteristics of the composition and interactions in the full-scale offshore oil-produced wastewater treatment plant.

## Materials and methods

### Operation of the offshore oil produced wastewater treatment plant

The produced wastewater treatment plant is a part of a terminal factory located in Guangxi province, China. Produced oil-water fluids from offshore oil production

platforms are transported through submerged pipelines to terminal factories, where produced oil and gas are processed to qualified products and produced wastewater is treated and discharged according to Chinese water discharge standards. The samples were collected by Linhai Wang who was employed by CNOOC Energy Development Co. Ltd. Technology Branch, and permitted by the CNOOC Energy Development Co. Ltd (No. GC2018ZCYH0107).The full-scale operating schematic diagram of the wastewater plant is shown in Fig 1. Wastewater from three separators is settled in a horizontal oil eliminator with an upper oil slick, and then the lower water phase is filtrated through a vertical walnut shell filter and flows into a settlement pond (SP). These processes are followed by downstream biological treatment processes with a maximum treatment capacity of 1000 m$^3$/day. Wastewater flows into an ABR and is treated for 12 hours; then the effluent from the ABR is treated in three APRs (APR1, APR2 and APR3) for 12 hours. Finally, treated water is discharged to sea.

Samples from an ashore oil-water (AOW), SP, ABR, and three aerobic parallel reactors (APR1, APR2, and APR3) were collected. COD was determined using the potassium dichromate oxidation method (high-chlorine wastewater determination of chemical oxygen demand–chlorine HJ/T 70–2001). NH$_3$-N was measured with a detector (5B-3C, Hualian Corporation, China). Oil components in the wastewater were determined using the national standard method (GB/T16488-1996). The concentrations of Cl$^-$ and SO$_4^{2-}$ were detected with an

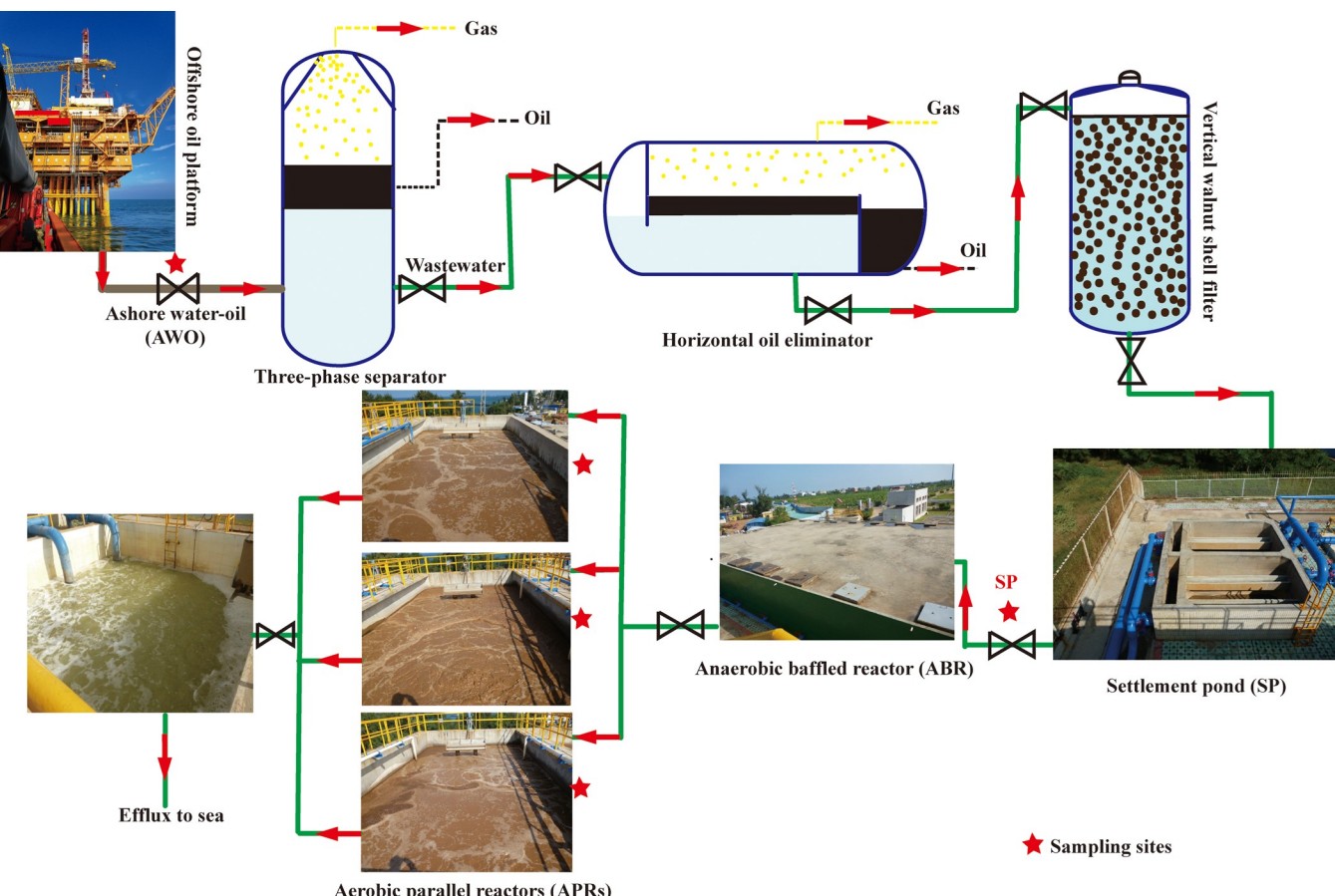

**Fig 1. A schematic overview of the full-scale offshore produced water treatment plant, red stars showing the sampling sites.**

ICS-600 ion chromatograph (Dionex, USA), and volatile fatty acids (VFAs) were detected with an ICS-2000 ion chromatograph (Dionex, USA).

## Pyrosequencing and bioinformatics analysis

For water samples, approximately 500 mL of liquid was centrifuged at 10,000 ×$g$ for 10 min to pellet cells. Genomic DNA was extracted from the collected cells and sludge according to instructions of the Fast DNA SPIN kit for soil (Mp Biomedicals, USA).

The 454 pyrosequencing method was used for the 16S rRNA gene. The DNA template was amplified with primer pair 27F (5′–AGAGTTTGATCCTGGCTCAG–3′) and 533R (5′–TTAC CGCGGCTGCTGGCAC–3′) for the V1-V3 region of the bacterial 16S rRNA gene. The 533R primer included a Roche Lib-L adapter and sample-specific barcode (454 Life Science). PCR mixtures (50 μL) contained each primer (0.6 μM), approximately 5 ng of template NDA, 1× PCR buffer, and 2.5 U of DNA polymerase (MBI. Fermentas, USA) and had three replicates. The amplification program was carried out with a 4 min initial denaturation (94˚C), followed by 25 cycles of 94˚C for 30 s, 55˚C for 30 s, 72˚C for 30 s, and a final step of extension of 72˚C for 10 min. After the triplicate PCR products were pooled and purified using a DNA gel extraction kit (Axygen, China), amplicon pyrosequencing was performed from the A-end with a 454/Roche A sequencing primer kit on a Roche Genome Sequencer GS FLX Titanium platform (BGI Genomics Co., Ltd, Shenzhen, China).

Raw multiplexed sequence reads were processed according to QIIME-1.6.0 (www.qiime. org). In this study, the number of valid representative OTUs was excessively high for the construction of a phylogenetic tree, and thus OTUs (reads ≥ 1%) were used in constructing phylogenetic trees with the neighbor-joining method.

## Functional annotation of microbial taxa

The potential microbial functions and microbial interactions were profiled according to on available FAPROTAX database and were complemented with extracted relevant information from NCBI BLAST (https://blast.ncbi.nlm.nih.gov/Blast.cgi). Details of FAPROTAX is available at www.zoology.ubc.ca./louca/FAPROTAX. An uncultured member from the Roche Genome Sequencer GS FLX Titanium platform in the study was affiliated with a particular metabolic function when all cultured species or genera within the taxa had been identified to that function. By using a versatile script in FAPROTAX, prokaryotic taxon abundance profiles can be converted into putative functional group abundance profiles. Throughout the paper, unless otherwise mentioned, the OTU-based relative abundances of functional groups in each sample were provided with respect to the total number of reads in that sample. The FAPRO-TAX database is mainly derived from marine environments [13], which ensures its suitability in annotating potential functions for the offshore oil-produced wastewater samples.

## Results and discussion

### Pollutant removal efficiency

During full-scale treatment processes, water quality gradually reached the wastewater discharge standard (Table 1). The COD in the water phase of the AOW was 879 mg/L. The COD removal efficiency was 9.4% for physical processes and was 41.96% for anaerobic biological process. After aerobic biological treatment, the COD in the water phase of each aerobic reactor was below 100 mg/L (Table 1). $NH_3$-N concentration in the water phase of the AOW was 125.58 mg/L, decreased to 56.72 mg/L after the anaerobic treatment, and was below 1 mg/L in the effluent after aerobic treatment (Table 1). The COD and $NH_3$-N removal rates in our

**Table 1. Summary of water quality, numbers of reads and OTUs in samples of the full-scale offshore produced water treatment plant.**

| Parameter | Samples | | | | | | |
|---|---|---|---|---|---|---|---|
| | AOW | SP | ABRS | ABRW | APR1 | APR2 | APR3 |
| pH | 7.8 | 8.0 | 7.1 | 7.3 | 6.9 | 7.3 | 7.6 |
| COD$_{cr}$ (mg/L) | 879 | 796 | /[a] | 462 | 92 | 89 | 78 |
| NH$_3$-N (mg/L) | 125.58 | 122.79 | / | 56.72 | 0.81 | 0.92 | 0.49 |
| Oil (mg/L) | | 48 | / | 15 | 4 | 2 | 2 |
| Cl$^-$ (mg/L) | 14826 | 13978 | / | 14057 | 13696 | 13645 | 13934 |
| SO$_4^{2-}$ (mg/L) | 87.56 | 42.75 | / | 169.8 | 141.3 | 156.3 | 136.5 |
| Acetate (mg/L) | 676.11 | 165.78 | / | 397.23 | 13.61 | 8.64 | 6.50 |
| Propionate (mg/L) | 19.55 | 22.63 | / | N/A | N/A[b] | N/A | 4.30 |
| Butyrate (mg/L) | 56.12 | 43.32 | / | 10.96 | N/A | N/A | N/A |
| No. reads | 10816 | 10810 | 10015 | 6405 | 6522 | 9922 | 11843 |
| No. OTUs | 728 | 324 | 1216 | 920 | 937 | 1264 | 1530 |

[a] It was not detected as there was not too much water phase in the sample of ABRS.

[b] Below the detected limitation.

anaerobic and aerobic reactors were inconsistent with those in previous studies, in which anaerobic reactors were mostly used in converting COD and aerobic reactors were efficient in removing organic nitrogen [15]. Li et al. [2] studied an anaerobic biofilm reactor capable of 95% COD removal for oily wastewater. Until now, factors influencing the performance of biological processes have attracted considerable interest [16]. Variations in biological wastewater treatment processes depend on many factors, such as local conditions, influent characteristics, reactor design, and operational parameters. In contrast to laboratory-scale reactors, a full-scale plant can remove unique pollutants and microbial characteristics because of the distinctive chemical and biological nature of offshore oil-produced wastewater.

The concentration of Cl$^-$ remained relatively constant in the full-scale steady treatment system and had a high range (13696–14826 mg/L), which was consistent with the Cl$^-$ concentration in the local seawater and implied the high salinity of the wastewater. The main type of VFAs in the AOW was acetate, which had a concentration of 676.11 mg/L. Butyrate and propionate were also detected at lower concentrations (56.12 and 19.55 mg/L, respectively). After the physical processes the concentration of acetate dropped sharply to 165.78 mg/L, the concentrations of the other VFAs only showed slight changes. In the water from the ABR, the concentration of acetate fluctuated to 397.23 mg/L, and the concentrations of the VFAs were low after the aerobic process (Table 1). All the monitored parameters indicated that the full-scale treatment system had an efficient performance during our study.

## Microbial composition is developed from oil- and marine-associated environments

Seven bacterial communities in the samples of AOW, SP, ABR sludge (ABRS), ABR water (ABRW), sludge of the three aerobic parallel reactors (APR1, APR2, and APR3) were obtained through 454 pyrosequencing. A total of 66,333 valid sequences were detected and clustered into 5333 species-level (97% similarity) OTUs aligned to be bacteria. The entire set of the raw sequences is available at ENA's Sequence Read Archive under accession number PRJ383980 (SRR5482389-SRR5482395).

Dominant microbial distributions at phylum level are shown in Fig 2. The highest diversity richness was observed in the samples from biological treatment processes in APR3, with 1530

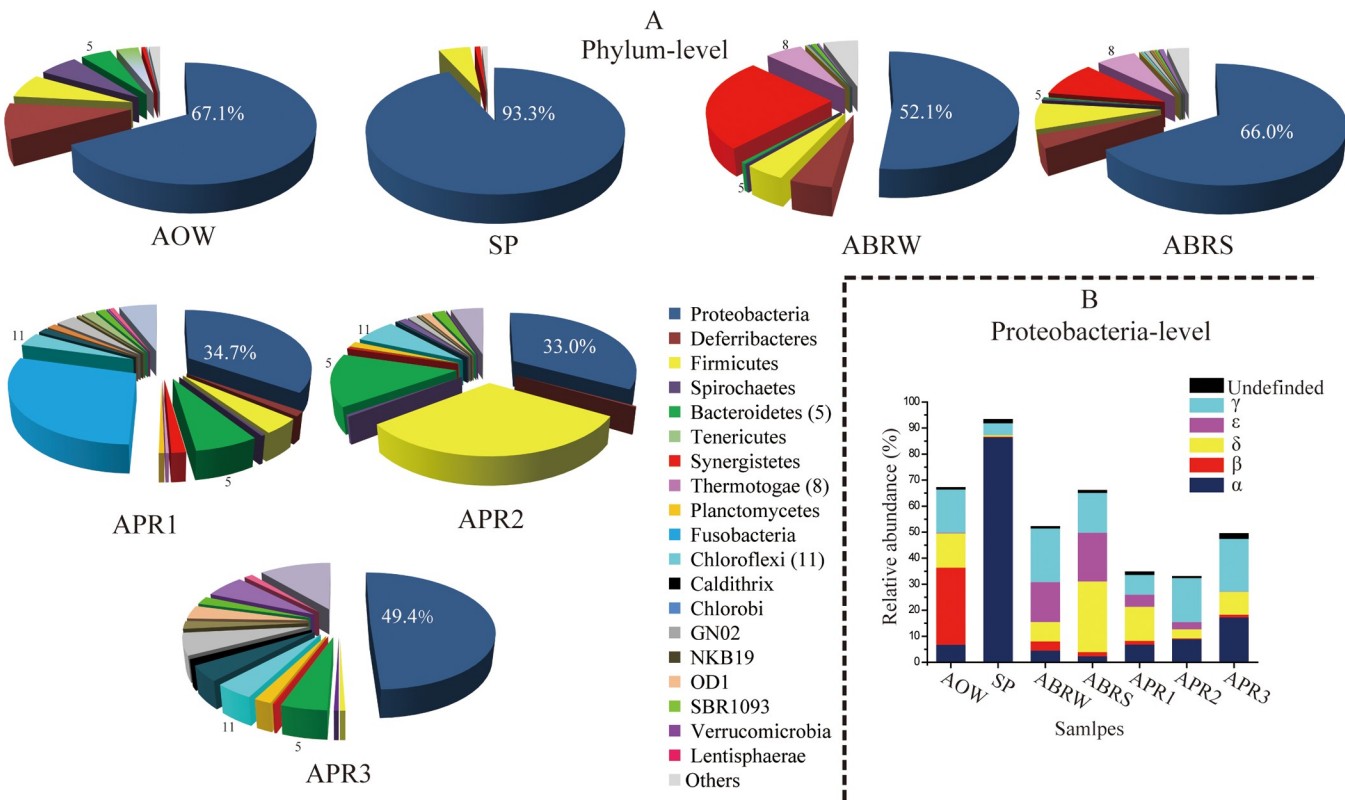

**Fig 2. Pie charts showing the dominant distributions of the detected microbes at the level of phylum in the collected samples; histograms showing the dominant distributions of detected proteobateria-affiliated microbes at the level of class in the collected samples.**

OTUs. The sample obtained through physical treatment processes showed the lowest diversity, with 324 OTUs in SP (Table 1). The functional annotation of the OTUs was conducted on the basis of FAPROTAX. A total of 1312 OTUs (24.6% of total OTUs) were identified to 63 functional assignments. The functional distributions of dominant function groups in each sample are illustrated in Fig 3.

We blasted each dominant OTU (relative abundance ≥ 1%) with the NCBI database. A majority of the detected sequences in the oil-produced wastewater treatment plant showed the closest similarities to the sequences detected either in oil reservoirs or in marine-associated samples (Figs 5 and S1 and S2 ). Recently, researchers have announced the significance of revealing microbial diversity and biogeography of bacterial communities in wastewater treatment plants and highlighted the importance of microorganisms in waste water treatment system (WWTS) are essential for water purification [12]. Using data from 269 WWTS in 23 countries on six continents, Wu et al. [12] reported that activated sludge microbiomes are mostly freshwater populations. Guo et al. [17] assessed the possible origins of wastewater microorganisms, including human feces and soil microorganisms. Ibarbalz et al. [18] observed that each industrial activated sludge system exhibits a unique bacterial community composition. However, the characteristics of microbial origin and composition of oil-produced wastewater remain poorly understood.

Microbial communities in inhabited environments reflect the selective pressure imposed by environmental impactors [19]. In particular, oil-produced wastewater, especially that from offshore oil fields, is highly distinctive in terms of chemical and biological nature; it is enriched with complex dissolved and dispersed oil components and has a high salt content, limited

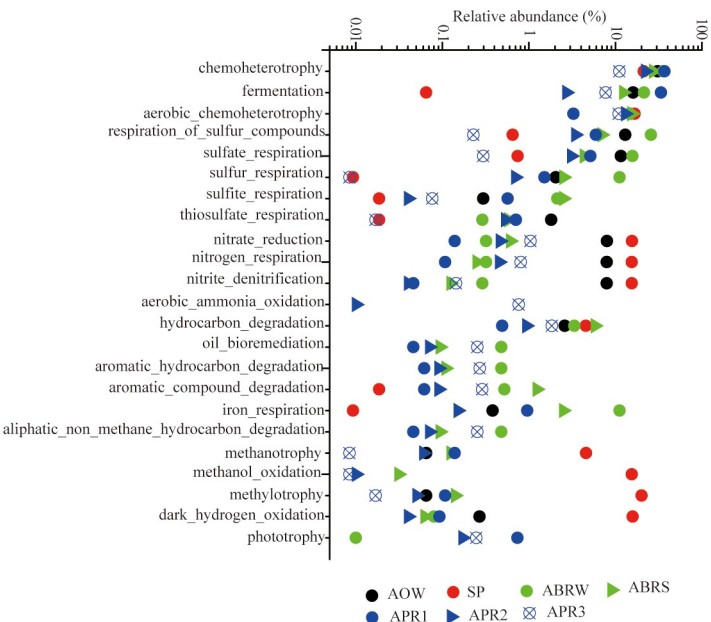

**Fig 3. Plots of relative abundances of OTUs associated with various dominant metabolic functions annotated by FAPROTAX software.**

nitrogen, and high sulfate content [20, 21]. Consistently, most of the detected sequences in the wastewater treatment system showed closest similarity to sequences detected either in marine environments or in oil-associated environments, demonstrating that the microbes that accumulated in the wastewater treatment plant had a common regional origin. Unlike information about microbial composition in municipal wastewater treatment plants investigated in many studies, information about the microbial composition characteristics, especially microbial origin, of oil-produced wastewater is limited. The data blasted with NCBI database are the first observations reported in the literature about microbial origin in oil-produced wastewater treatment systems.

## Physical and chemical treatments system resulted in low microbial diversity

The microbial community of the AOW dominantly consisted of bacteria affiliated within the genera *Desulfovibrio*, *Flexistipes*, *Pseudomonas*, *Novispirillum*, *Halanaerobium*, *Sphaerochaeta*, *Acholeplasma*, *Marinobacterium*, and *Marinobacter*, and Rhodocyclaceae (S1 Fig). Microbes in the AOW showed great response to upstream physical and chemical treatments. First, microbial diversity in the SP declined as the number of detected OTUs decreased sharply from 728 in the AOW to 324 in the SP. Second, α-proteobacteria constituted 86.70% of the total reads in SP and had only the two dominant genes of *Tistrella* (58.70%) and *Paracoccus* (12.64%; S1 Fig). The results indicated that the upstream physical and chemical treatments resulted in a low level of microbial diversity.

Microbial functions in AOW and SP also showed great differences. Of the considered functional groups, chemoheterotrophy, fermentation, respiration of sulfur compounds, nitrate reduction, and hydrocarbon degradation were annotated to be dominant in the AOW (Fig 3). In the SP, the functional groups of fermentation and respiration of sulfur compounds were below 1%, whereas the functional groups of methylotrophy, methanol oxidation, dark

hydrogen oxidation, and methanotrophy were dominant (Fig 3). The metabolic functions of sulfur compounds and fermentation were inhibited in the SP, whereas bacteria with the ability to metabolize compounds devoid of carbon–carbon bonds became predominant.

The low level of microbial diversity and function changes after upstream treatment might be attributed to the addition of chemicals that are typically used to kill harmful microbes, such as sulfate-reducing bacteria (SRB), demulsify oil–water emulsion, and settle solid particles [22]. Most SRBs present in oil fields belong to δ-aproteobacteria and Clostridia [23]. The Deltaproteobacteria SRBs of *Desulfovibrio* and Clostridia SRBs of *Halanaerobium* and *Fusibacter* were dominant in the offshore oil-water sample (AOW). The addition of commercial bactericides mainly resulted in the disappearance of the dominant SRBs OTU 3101 (*Desulfovibrio*), OTU 6337 (*Desulfovibrio*), OTU 1740 (*Halanaerobium*), and OTU 3568 (*Fusibacter*) in the SP (S1 Fig). Fermentation seems to be the dominant process degrading organic matter in oil reservoirs [24]. However, in the SP, the microbial function of fermentation declined sharply to 0.065%. Except the direct influence of bactericides on the growth of fermenters, decline in the fermentation function in the SP might be associated with the inhibition of sulfur compounds' respiration functions. Given that fermentation products are the main substrates of SRBs in oil reservoirs environments [24], the inhibition of SRBs resulted in the accumulation of the products of fermenters and may have affected the fermentation microorganisms' ability to grow [25]. Moreover, the eugenic growth of methylotrophic α-proteobacterium that can utilize toxic one-carbon compounds, such as N, N-dimethylformamide, formamide, methanol, and methylamines [26], implies the environmental pressure of bactericides on microbial community in the upstream treatment system.

Normally, upstream treatment system for oil-produced wastewater aims to reduce suspended solids and dispersed oil through physical and chemical methods. Small emphasis has been put on the response of microbial community during the processes. Shifts in microbial communities and functions during upstream treatments indicated that the predominant microbial communities survived and highlighted the indicators of chemicals discharged to the environment.

## Fermentation based on sulfur compound respiration for organic pollutant removal in the anaerobic baffled reactor

Unraveling the microbial ecology in biological treatment systems is always the first and the most significant step for understanding the in the situ bioprocesses [27]. Anaerobic digestion is a promising strategy for treating wastewater in upflow anaerobic sludge blanket reactors and anaerobic membrane bioreactors. We focused on microbial communities and functions in the sludge phase (ABRS) and water phase (ABRW) in the operating anaerobic baffled bioreactor to reveal intensive ecological interrelationships.

Microbial communities in ABRS and ABRW showed a close relationship with 276 shared OTUs and constituted 70.27% and 69.02% of the total reads, respectively. The main differences between the two samples were the microbial relative abundances in the phylum of δ-, γ-proteobacteria and Synergistetes. The ABRS had 27.13% of δ-proteobacterial reads, whereas the ABRE only had 7.40% of δ-proteobacterial reads (Fig 2). Compared with the coverage of 9.67% Synergistetes-affiliated microbes in the ABRS, Synergistetes-affiliated microbes covered the highest relative abundance of 25.26% in the ABRW (Fig 2). The functional annotations of OTUs in the ABRS and ABRW showed great similarity with dominant fermentation, respiration of sulfur compounds, hydrocarbon degradation, and iron respiration (Fig 3).

In the oil field-produced wastewater, COD contributors included soluble hydrocarbons, which were mainly benzene, toluene, ethylbenzene, and xylene isomers; emulsified insoluble

hydrocarbons; organic surfactants; and polymers [20, 28]. The bioconversion of organic matter in the ABR was associated with anaerobic fermentation. In the detected microbial community, hydrolysis and acidogenesis processes seemed to be activated in the ABR because Clostridiaceae and Thermotogae-affiliated OTUs were dominant in the ABR samples. Previous studies pointed out that many species of Clostridiaceae have strong hydrolytic ability [29] and Thermotogae can produce hydrolytic enzymes that catalyze the breaking down of various polysaccharides to acetate, hydrogen, and carbon dioxide [30]. Fermentation commonly occurs in anaerobic bioreactors, and the key step in the biotransformation of organic micropollutants is acidogenesis [31]. VFAs are the major intermediate metabolites during anaerobic digestion [32].

Another process involved in the bioconversion of organic matter in the ABR was initiated with hydrocarbon-degradation. γ-Protobaceria-affiliated *Marinobacterium*, *Marinobacter*, *Pseudomonas*, and *Halomonas* were dominant in ABRS and ABRW. Nearly all the sequences of the γ-protobaceria-affiliated OTUs showed the closest similarities to the sequences detected from oil-related environments (S2 Fig) and possess the ability to degrade hydrocarbons or polymers [33, 34]. Another main group of hydrocarbon-degrading microbes were δ-protobaceria-affiliated *Desulfotignum* and *Desulfoglaeba*. *Desulfotignum*-like OTU 4339 was closely affiliated with cultured *Desulfotignum toluenicum* strain H3(T), which oxidizes toluene coupled to hydrogen sulfide production [35], *Desulfoglaeba*-like OTU 2021 showed 99% similarity to cultured *Desulfoglaeba alkanexedens* strain ALDC, which can oxidize n-alkanes completely [36].

Electron acceptors play a significant role in regulating pollutant bioconversion. The main electron acceptors include $O_2$, $NO_x^-$, and $SO_4^{2-}$ [37]. γ-Protobaceria-affiliated *Marinobacterium*, *Marinobacter*, *Pseudomonas*, and *Halomonas* represent aerobic-chemoheterotroph function [33], and $O_2$ is the main electron acceptor during hydrocarbon and polymer degradation. $SO_4^{2-}$ is the electron acceptor for δ-protobaceria-affiliated *Desulfotignum* and *Desulfoglaeba* [35]. The results implied that aerobic and anaerobic hydrocarbon and polymer-degradation occurred in the ABR. Unlike in laboratory-scale treatment, completely preventing $O_2$ import in a field trial treatment plant is difficult because the inlet process does not involve a pretreatment process for $O_2$ removal. $O_2$ import has no adverse effect on organic component removal but can enhance anaerobic biodegradation efficiency through the hydrolytic acidification process [38, 39]. The dominant γ-protobaceria-affiliated *Marinobacteriu*, *Marinobacter*, *Pseudomonas*, and *Halomonas* degrade hydrocarbons through the intracellular attack of oxygenases and perxidases [33, 40]. Therefore, the detected γ-protobaceria-affiliated microbes played a main function in the aerobic degradation of hydrocarbons.

Apart from the end products of $CO_2$ and $H_2O$, hydrocarbon degradation and hydrolytic acidogenesis generate a substantial amount of intermediate organic acids, especially acetate. The lack of acids utilizers would result in the accumulation of metabolic products and decline in pollutant removal efficiency of an ABR [30]. According to the data of detected microbial communities in the ABR, intermediate product metabolism had two potential pathways. The first one was associated with organic acid utilization by the groups of Synergistetes microbes that were able to perform acetogenesis or interact syntrophically with methanogens [30, 41]. The Synergistetes-affiliated microbes were dominant in the ABRW (25.26%) and ABRS (9.67%) (Fig 2) and belonged to the groups of *Anerobaculum*, *Thermovirgaceae*, and *Dethiosulfovibrionaaceae*. The Synergistetes-affiliated microbes are frequently detected in environments associated with petroleum reservoir fluids and convert organic acids into acetate, hydrogen, and $CO_2$ [42].

The second path for intermediates, especially acetate, was associated with the sulfur cycle involving sulfate reduction to sulfide ($SO_4^{2-} \rightarrow S^{2-}$) and subsequent sulfide oxidation to

elemental sulfur ($S^{2-} \rightarrow S^0$), which always occur in wastewater biofilm or activated sludge [43]. Function announcement showed that 25.59% and 6.96% of the detected microbes in ABRS and ABRW, respectively, had functions for the respiration of sulfur compounds (Fig 3). In the ABR, apart from the hydrocarbon-degrading SRBs of *Desulfotignum* and *Desulfoglaeba*, a group of sulfate utilizer was affiliated with *Desulfobacter* (Fig 4). *Desulfobacter*-like bacteria are strict anaerobes that use sulfate as their usual terminal electron acceptor and are complete oxidizers that oxidize acetate to $CO_2$ [44]. The formation of sulfide upon the reduction of sulfate always is one of the problems associated with anaerobic wastewater treatment. Sulfide is removed by the direct introduction of air into anaerobic bioreactor systems or biological conversion of sulfide to elemental sulfur ($S^0$) using sulfide-oxidizing bacteria (SOB) with $O_2$ and $NO^{3-}$ as electron acceptors [43]. In the ARB, OTU 271 affiliated with Campylobacterales were dominant in ABRS (15.63%) and ABRW (20.79%). Campylobacterales are mostly sulfide oxidizers commonly detected in marine and oil reservoirs, and $NO^{3-}$ is the electron acceptor [45]. The existence of dominant *Desulfuromonas*, a common genus of sulfur-reducing bacterium using acetate as an electron donor [30, 46], implied that SOBs converted sulfide to elemental sulfur ($S^0$) in the ABR. Furthermore, the abundance of *Desulfuromonas* in the ABRS (11.08%) was higher than that in the ABRW (2.48%; Fig 4), implying that biological sulfide oxidation resulted in the precipitation of sulfur ($S^0$) in sludge phase. The microbial communities

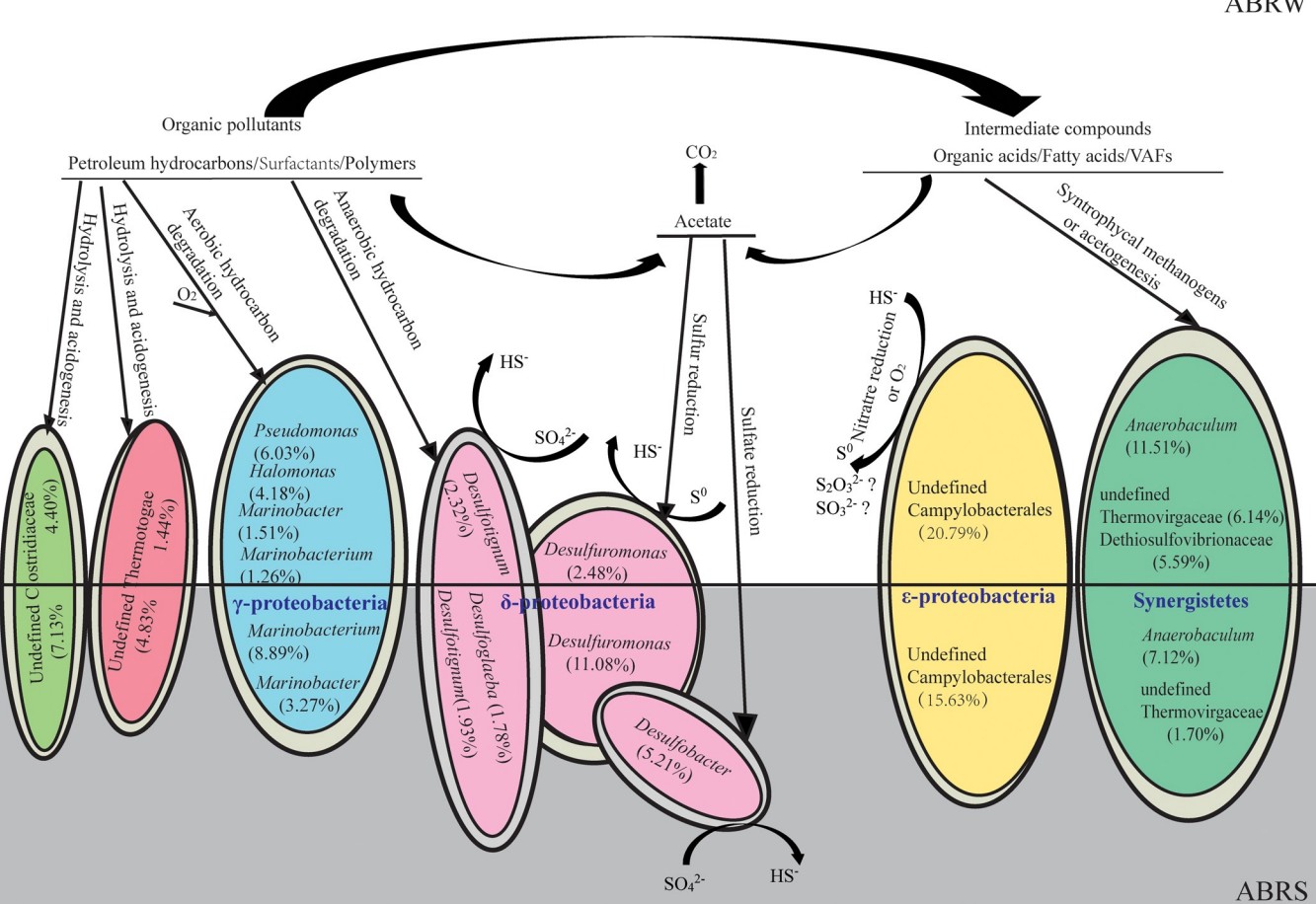

**Fig 4. Schematic diagram of the ecosystem involving dominant microbial distributions and potential organics conversion processes in the water phase (ABRW) and sludge phase (ABRS) of the anaerobic baffled reactor.**

in the ABR performed an apparent sulfur cycle that contributed to COD removal, which is consistent with the application of sulfur-mediated biological wastewater treatment systems [47]. In environments with high concentration of sulfates, SRBs use a wide range of carbon sources to produce hydrogen sulfide ($H_2S$), and the removal of $H_2S$ promotes rapid organic removal [47]. Consistently, sulfur compound respiration in the ABR was reasonable because of the high sulfate content in the offshore oil-produced wastewater.

Dominant microbial communities and organics conversion process are illustrated in Fig 4. The aerobic and anaerobic degradation of hydrocarbons and fermentation-based hydrolysis acidogenesis intensified the biodegradability of organic pollutants to lower molecular substrates. Inconsistent with our expectation that anaerobic hydrocarbon degradation would be dominant in the anaerobic reactors, aerobic hydrocarbon degraders contributed mainly to hydrocarbon degradation in the ABR. This result implied that the anaerobic baffled reactor was an environment with a gradient of oxygen concentrations. Given that reports on in situ microbial communities for oil-produced water biotreatment systems are few, we concluded that anaerobic baffled reactors could act as aerobic and anaerobic co-existing chemostats. Aerobes and anaerobes co-existing in ecology systems are widely distributed in natural environments and can significantly enhance pollutant removal [38, 39, 48]. Therefore, the co-located anaerobic aerobic microbes in our anaerobic baffled reactor were reasonable and worked efficiently in removing of oil and COD (Table 1). Although sulfur compound respiration and syntrophical methanogens or/and acetogenesis consumed a large part of acetate in the ARB, the concentration of acetate in the ARB increased to 397.23 mg/L, and the COD remained 462 mg/L (Table 1). Normally, most previous studies pointed out that although anaerobic reactors are efficient in removing biodegradable organics, their effluent quality often does not meet the permissible limits for discharge. Therefore, an adequate post-treatment system is necessary for further removing remaining organic pollutants.

## Different microbes play similar functions in the three aerobic parallel reactors

After anaerobic treatment, an aerobic treated system with three aerobic parallel batch reactors (APRs) was used in the produced water treatment plant. Microbes inhabiting the activated sludges of the three aerobic reactors were investigated. The three aerobic reactors were in parallel and filled with wastewater from the same batch of the anaerobic baffled reactor in this study. Unexpectedly, microbial communities in the three parallel reactors showed significant differences. Apart from the different distributions in proteobacteria-affiliated microbes, Fusobacteria-affiliated microbes were dominant in APR1, Firmicutes-affiliated microbes were dominant in APR2, and groups with unknown candidate divisions were dominant in APR3 (Fig 2). When the FAPPOTAX database was used in annotating microbial functions in the three APRs, we were unable to obtain consistent and reasonable function annotations because most of the detected dominant OTUs were affiliated with bacteria below the order level (Fig 5). To reveal the characteristics of the microbial functions in the three APRs, we blasted the sequences of dominant OTUs (relative abundance ≥1%) with the NCBI database to elucidate potential function groups. According to the detail analysis of the blast results, dominant microbes in the APRs seemed to play similar functions.

First, aerobic hydrocarbon-degrading microbes were dominant in the three APRs. The members of γ-proteobacteria, commonly found during hydrcarbon degradation, comprised 20.28% in APR3, 16.87% in APR2, and 7.60% in APR1 (Fig 2). *Marinobacter-*, *Pseudomonas-*, *Enterobacter-*, and Alcanivoracaceae-like microbes were dominant in APR3; *Halomonas-*, *Idiomarina-*, and nsmpVI18-like microbes were dominant in APR2; and *Marinobacterium-*

like microbes were dominant in APR1(Fig 5). The γ-proteobacteria microbes always responded faster to hydrocarbon-associated environments [49], suggesting that they potentially played an important role in hydrocarbon depletion in the aerobic reactors.

Second, Microbes possessing the acidogenesis function were dominant. In APR1, dominant acidogenesis OTUs were affiliated with Fusobacteria (29.00%), Saprospiraceae (3.93%), and OD1 (1.98%). In APR2, dominant acidogenesis OTUs were Bacteroidetes-affiliated microbes (14.42%), *Anaerobacillus* (11.37%), *Clostridium* (8.06%), and OD1 (3.69%). In APR3, dominant acidogenesis OTUs were affiliated with Verrucomicrobia (6.18%), Caldithrix (4.29%), and Clostridia (1.36%) (Fig 5). Acidogenesis is the second step of anaerobic fermentation. Detected *Propionigenium*-like microbes (Fusobacteria) are potentially secondary fermenters that preferably use decarboxylic acids as substrates in their fermentative metabolism to produce propionate [50]. Firmicutes-, Bacteroidetes-, Verrucomicrobia-, Caldithrix-, and OD1-affiliated microbes are the most common found phyla during acidogenesis in bioreactors [30, 32, 51, 52]. According to the blast results obtained using the NCBI database, all the above representative OTUs were affiliated with reported anaerobic microbes (Fig 5). Consistently, it is usual to detect microaerophilic to obligate anaerobic microbes in aerobic wastewater treatment systems, as organic matter from wastewater is initially adsorbed to an activated sludge and degraded by microbes inhabiting the inner phase of the sludge [53].

Third, VFAs produced by hydrocarbon-degradation and acidogenesis processes can be further converted by photofermentation biohydrogen-producing and nitrogen-removing microbes in the three APRS. Detected microbes with potential photofermentation function were GN02-, Rhodospirillaceae-, and Rhodobacteraceae-affiliated microbes. GN02-affiliated OTUs occupied 3.07% (APR1), 6.03% (APR2), and 1.31% (APR3), and Rhodospirillaceae and Rhodobacteraceae-affiliated microbes occupied 2.71% (APR1), 3.01% (APR2), and 2.85% (APR3). Detected GN02-affiliated OTUs showed the closest similarity to microbes in Guerrero Negro hypersaline microbial mat and yielded novel photosynthetic pathways for energy assimilation [54]. Rhodospirillaceae and Rhodobacteraceae-affiliated microbes are common photosynthetic hydrogen producing microbes that convert organic wastes to hydrogen energy through photofermentative processes [55], which have been assessed for wastewater treatment. However, most studies were focused on hydrogen production rather than the removal of organics [56]. The high COD removal rates and dominant photofermentation biohydrogen-producing microbes in the APRs indicated that photofermentative processes coupled with acidogenesis fermentation are promising and efficient strategies for organic pollutant removal in oil field-produced wastewater. Combining dark fermentation and photofermentation processes enhances biohydrogen production [57]. The detected microbes that had potential biological nutrient nitrogen removal capabilities were Chloroflexi-, and Hyphomicrobiaceae-affiliated microbes. Chloroflexi-affiliated microbes had percentages of 6.53% in APR2, 4.31% in APR3, and 3.80% in APR1(Fig 2). Hyphomicrobiaceae-affiliated OTUs comprised 1.6%, 5.4%, and 1.6% of the total reads in each library. The members of the phylum Chloroflexi are filamentous bacteria in activated sludge wastewater treatment plants and commonly found in treatment plants designed to remove nitrogen [58]. Hyphomicrobiaceae-affiliated microbes in wastewater systems typically possess nitrogen-removal enzymes and are closely related to nitrogen removal [59–61].

Although the diversity of microbes in the three aerobic reactors showed great differences, their functions appeared consistently similar. Previous studies on aerobic activated sludge ecosystems showed that microbial populations can vary significantly without affecting treatment efficiency [62]. Each reactor of our aerobic biologic treatment system harbored hydrocarbon degraders, complex organic compound fermenters, photosynthetic assimilators, and nitrogen removers. These microbes formed their unique ecological systems with different microbial

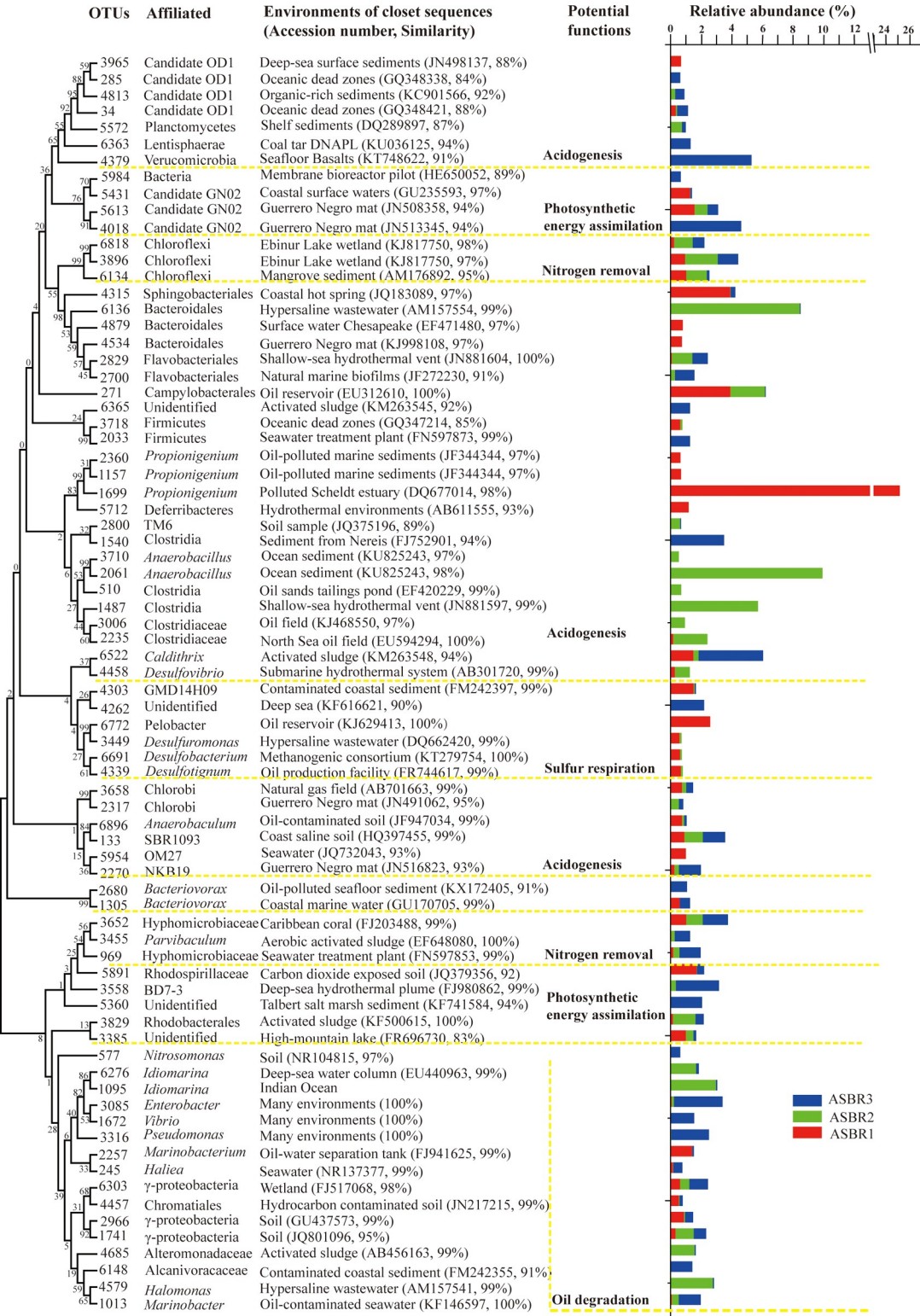

**Fig 5. Bacterial phylogenetic tree of the dominant OTUs (relative abundance ≥0.5%) in the three aerobic parallel reactors constructed by the neighbor-joining method.** Potential function groups inferred from the relevant information extracted from NCBI database. Histograms showing the relative abundances (%) of reads of each dominant OUT detected in the three samples.

distributions in each reactor and cooperated successfully in removing organic matter in the wastewater. This study thoroughly investigated microbial communities in an aerobic offshore oil-produced water treatment system. However, as the microbial communities of the studied reactors were complex and consisted of a large percentage of uncultured candidate divisions, drawing the potential metabolize network in each reactor was difficult. New techniques such as long-read sequencing methods are being increasingly applied to identify microbial community members in wastewater treatment plants [63, 64].

## Conclusions

In this study, we characterized the composition and potential functions of the bacteria inhabiting in a full-scale offshore oil-produced wastewater treatment plant. Owing to the distinctive nature of offshore oil-produced water, microbes dominant in the plant originated mostly from oil and marine environments. Although the upstream physical and chemicals processes caused low microbial diversity, microbes in the anaerobic and aerobic reactors developed effective interactions for pollutant removal. Fermentation and sulfur compound respiration performed a unique microbial network in the ABR. Diverse microbes possessed similar functions, such as hydrocarbon degradation, acidogenesis, photosynthetic assimilation, and nitrogen removal, in the APRS. To the best of our knowledge, this study is the first to explore microbial assemble patterns with various potential metabolic functions in an offshore oil-produced wastewater treatment plant and can provide a rather comprehensive ecology and new insight into the characteristics of bacterial communities relevant to the efficiency of oil-produced wastewater treatment plants.

## Supporting information

**S1 Fig. The dominant OTUs (reads> 1%) alignment analysis of the samples collected from ashore water-oil (AWO) and settlement pond (SP).** The bacterial phylogenetic tree of the dominant OTUs was constructed using the neighbor-joining method. Scatter points showing the relative abundances (%) of reads for each dominant OUT detected in the three samples. Samples were shown in different colors.
(TIF)

**S2 Fig. The dominant OTUs (reads> 1%) alignment analysis of the samples collected from anaerobic baffled reactor (ABRW and ABRS).** The bacterial phylogenetic tree of the dominant OTUs was constructed using the neighbor-joining method. Scatter points showing the relative abundances (%) for reads of each dominant OUT detected in the three samples. Samples were show in different colors.
(TIF)

**S1 Graphical abstract.**
(TIF)

## Author Contributions

**Conceptualization:** Yuehui She, Fan Zhang.

**Data curation:** Shuyuan Deng, Bo Wang, Sanbao Su.

**Formal analysis:** Shuyuan Deng, Fan Zhang.

**Funding acquisition:** Yuehui She, Fan Zhang.

**Investigation:** Shuyuan Deng, Bo Wang, Wenda Zhang.

**Methodology:** Wenda Zhang, Shanshan Sun, Weiming Liu.

**Project administration:** Yuehui She, Fan Zhang.

**Resources:** Linhai Wang.

**Software:** Jianping Guo.

**Supervision:** Yuehui She, Fan Zhang.

**Validation:** Ibrahim M. Banat, Yuehui She.

**Visualization:** Sanbao Su, Hao Dong.

**Writing – original draft:** Shuyuan Deng.

**Writing – review & editing:** Shuyuan Deng, Wenda Zhang, Ibrahim M. Banat, Fan Zhang.

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
