## [Decision Letter · Decision Letter 0]

27 Apr 2021

PONE-D-21-08773

Elucidate microbial characteristics in a full-scale treatment plant for offshore oil produced wastewater

PLOS ONE

Dear Dr. Zhang,

Thank you for submitting your manuscript to PLOS ONE. After careful consideration, we feel that it has merit but does not fully meet PLOS ONE’s publication criteria as it currently stands. Therefore, we invite you to submit a revised version of the manuscript that addresses the points raised during the review process.

Please find attached three reviews of your manuscript. All three reviewers were generally positive about your manuscript. At the same time, they raised concerns which need to be addressed, together with detailed comments for you to further improve the manuscript. After careful consideration and based on the reviewers' comments, we invite you to submit a revised version.  Further consideration of the manuscript will be contingent upon revision according to the detailed reviewers' suggestions.  

We look forward to receiving your revised manuscript.

Kind regards,

Guanglei Qiu

Academic Editor

PLOS ONE

Journal Requirements:

2. In your Methods section, please provide additional information regarding the permits you obtained to collect samples for the present study. Please ensure you have included the full name of the authority that approved the field site access and, if no permits were required, a brief statement explaining why.

3. Please include your tables as part of your main manuscript and remove the individual files. Please note that supplementary tables (should remain/ be uploaded) as separate "supporting information" files.

5. Thank you for stating the following in the Financial Disclosure section:

[This study was funded by National Natural Science Foundation of China (Nos. 51504221, 51774257, 51574038 and 51634008), and National Science and Technology Major Oil and Gas Special Project (2017ZX05009-004).].   

We note that one or more of the authors are employed by a commercial company: Sinopec Shengli Oilfield

6. Please include a caption for each figure in the manuscript and not as a separate file.

7. Please include captions for ALL your Supporting Information files at the end of your manuscript, and update any in-text citations to match accordingly. Please see our Supporting Information guidelines for more information: http://journals.plos.org/plosone/s/supporting-information.

Reviewers' comments:

Reviewer's Responses to Questions

**Comments to the Author**

1. Is the manuscript technically sound, and do the data support the conclusions?

Reviewer #1: Partly

Reviewer #2: Partly

Reviewer #3: Yes

2. Has the statistical analysis been performed appropriately and rigorously? 

Reviewer #1: N/A

Reviewer #2: No

Reviewer #3: Yes

3. Have the authors made all data underlying the findings in their manuscript fully available?

Reviewer #1: Yes

Reviewer #2: Yes

Reviewer #3: Yes

4. Is the manuscript presented in an intelligible fashion and written in standard English?

Reviewer #1: No

Reviewer #2: Yes

Reviewer #3: Yes

5. Review Comments to the Author

Reviewer #1: Ref: PONE-D-21-08773

Title: Elucidate microbial characteristics in a full-scale treatment plant for offshore oil produced wastewater

This manuscript elucidated the microbial composition and the potential microbial functions in a full-scale well-worked offshore oil produced wastewater treatment plant using 16S rDNA amplicon sequencing. In general, the finding of this work is interesting. However, the following comments should be addressed.

(1)The level of English should be improved and the data format should be unified throughout the manuscript.

(2)In “Highlight” sections, their content should be rewritten and simplified for better description the manuscript.

(3)If the abbreviation appears for the first time in the manuscript, please indicate its specific name. (like VFA, WWTS)

(4) “NH3-N concentration in the water phase of ashore water-oil sample was 125.58 mg/L, after the anaerobic treatment the NH3-N concentration decreased to 56.72 mg/L”. Please explain how NH3-N was removed in ABR.

(5) Concentration of sulfate was increased in ABR, why? How COD was removed in ABR? Why was not methane produced in ABR?

(6) What was the physical processes?

(7) The salt concentration in the oil production wastewater is high. In the process of biological treatment of wastewater, how much influence does it have on the treatment efficiency of microorganisms?

(8) In three aerobic reactors, the diversity and functional similarities of microorganisms in three different aerobic treatments need to be discussed in more detail.

(9) The layout of Figure 3 May be improve and some signs overlap so the signs are not clear.

(10) Please check “A majority of sequences detected by highthrough sequencing methods are identified with bacteria below to order lever that can not be affiliated with functional groups as many functions are only conserved at species or genus

level in FAPROTAX database. I”.

Reviewer #2: Review for PONE-D-21-08773

In this study authors describe the treatment of oil-produced wastewater. To improve treatment processes and meet stringent effluent discharge limits, it is important to identify the microbial members found in these systems and understand their metabolic potential. The topic is interesting, the experiments are comprehensive, the findings are useful, for adding in-depth understanding to biological wastewater treatment. To make it better, I have some suggestions/questions as below:

Pg 3, ‘…remains curtail to modern civilization..”

Sentence unclear

Pg 5, In this work, we elucidated the structures and potential functions of bacteria in a full-scale offshore oil produced wastewater treatment plant based on highthrough 16S rDNA sequences.

Do the authors mean 16S ribosomal RNA (rRNA) or recombinant DNA (rDNA)?

Pg 6, correct OIIME-1.6.0 Pipelines to QIIME-1.6.0…

Pg 8, ‘NH3-N concentration in the water phase of ashore water-oil sample was 125.58 mg/L, after the anaerobic treatment the NH3-N concentration decreased to 56.72 mg/L, and after aerobic treatment the NH3-N concentration in the effluent below 1 mg/L (Table 1).’

Ammonia usually has an inhibitory effect on anaerobic digestion but given that 334 mg/L COD reduction was observed, it might seem that the ammonia was either being stripped in gas phase or some microorganisms present in the digester could utilize ammonia since in section 3.4 there is mention of oxygen intrusion?

Pg 8, Section 3.2, It is important to statistically analyse changes in abundance of microbial community members (Fig 2) and metabolic functions (Fig 3) in different treatments and include it in discussions of sections 3.2 – 3.5.

Pg 9, ‘In addition, as seen in Fig. S1, only the rarefaction curve of AOW and SP tended to approach the saturation plateau. The rest of rarefaction curves show little sign of plateau, indicating that the biological treatment processes contained a somewhat broader phylogenetic diversity...’

α-, β diversity measures will be more conclusive than rarefraction curves. Rarefraction curves not plateauing may also refer to insufficient depth of sequencing.

Pg 12, ‘…which could provide microbial data to the downstream treatment system, especially to those systems using biotreatments.’

Sentence unclear

Pg 16, However, the aerobes and anaerobes co-existing ecology systems were reported popular in natural environments and could significantly enhance pollutants removal...

Pg 17, …as organic matters from wastewater are initially adsorbed to the activated sludge and are degraded by microbes inhabited in the inner of the sludge (Mario, et al., 2019).

Above explanations can be improved by discussing factors that lead to ecological gradients in sludge and how this allows microorganisms to co-exist in these conditions.

Pg 20 - Hyphomicrobiaceae-affiliated microbes inhabited in wastewater system were commonly detected to posse nitrogen removal enzymes and were closely related to nitrogen removal (Jia et al., 2019; Tomasek et al., 2017).

Authors may also discuss that Hyphomicrobium-related species are also commonly found in denitrifying wastewater treatment systems as identified in:

a) Lu, H., Chandran, K. and Stensel, D., 2014. Microbial ecology of denitrification in biological wastewater treatment. Water research, 64, pp.237-254.

Pg 20 – “However, as the microbial communities of the studied reactors were so complex and consisted of a large percentage of uncultured candidate divisions, it was not easy to draw the potential metabolize network in each reactor. More studies associated strains isolation and function genes detection are therefore important to follow up.”

Authors can mention that since many microorganisms could not be identified, it should be considered in future work on this topic. Further, strain isolation has several redundancies but new techniques such as long-read sequencing methods are being increasingly applied to identify microbial community members in wastewater treatment plants as shown in:

a) Singleton, C.M., Petriglieri, F., Kristensen, J.M., Kirkegaard, R.H., Michaelsen, T.Y., Andersen, M.H., Kondrotaite, Z., Karst, S.M., Dueholm, M.S., Nielsen, P.H. and Albertsen, M., 2021. Connecting structure to function with the recovery of over 1000 high-quality metagenome-assembled genomes from activated sludge using long-read sequencing. Nature Communications, 12(1), pp.1-13.

b) Arumugam, K., Bessarab, I., Haryono, M.A.S. et al. 2021. Recovery of complete genomes and non-chromosomal replicons from activated sludge enrichment microbial communities with long read metagenome sequencing. npj Biofilms Microbiomes 7, Nature Publishing Group, https://doi.org/10.1038/s41522-021-00196-6

Pg 20 – To increase the impact of this work and strengthen its relevance in the environmental context, the authors can briefly discuss the potential impact of climate change and rising temperature on the overall function of oil produced wastewater treatment plants since they are mostly inoculated with marine/oil sources. To assist the authors, there has already been some recent discussion regarding the impact of climate change on wastewater treatment and mitigation strategies, e.g.:

a) Lu, L., Guest, J.S., Peters, C.A., Zhu, X., Rau, G.H. and Ren, Z.J., 2018. Wastewater treatment for carbon capture and utilization. Nature Sustainability, 1(12), pp.750-758.

b) Qu, J., Wang, H., Wang, K., Yu, G., Ke, B., Yu, H.Q., Ren, H., Zheng, X., Li, J., Li, W.W. and Gao, S., 2019. Municipal wastewater treatment in China: Development history and future perspectives. Frontiers of Environmental Science & Engineering, 13(6), pp.1-7.

c) Qiu, G., Law, Y., Zuniga-Montanez, R., Lu, Y., Roy, S., Thi, S.S., Hoon, H.Y., Nguyen, T.Q.N., Eganathan, K., Liu, X. and Nielsen, P.H., 2021. Global warming readiness: Feasibility of enhanced biological phosphorus removal from wastewater at 35oC. bioRxiv.

Pg 20 – Conclusion: In this study, we have characterized the composition and interaction of the bacteria inhabiting in the full-scale…

>In my understanding, no interactions have been mentioned in the manuscript, rather the functional potential is discussed.

Table 1. Correct to AWO

Fig 1. Can mention in footnote that 2 samples, ABRS and ABRW, were collected from ABR.

Fig 2. Different colours to be used for alphaproteobacteria and unidentified taxa.

References – Please check the consistency of all references, including author names, journal names and titles.

Reviewer #3: Deng et al investigated the microbial ecology and microbial interactions in a full-scale offshore oil produced wastewater treatment plant. Results demonstrated that microbes inhabited in the plant were diverse and were originated from oil and marine associated environments, and the different microbial communities in three aerobic parallel reactors showed similar potential functions. This study is very interesting with a focus on the microbial communities and function in the offshore oil produced WWTPs, which could be a very special habitat. Overall, the manuscript is well written, however, the shortcomings of the methodologies (i.e., 16S rRNA sequencing and functional predilection) may limit the significance of this study. Major revisions are needed in view of the following comments before publishing.

BTW, there are no line numbers showing on the manuscript, which makes citing the location of comments need a bit problematic for the reviewer.

Major concerns:

-As you mentioned in Page 13 line 10-13, organic pollutants such as BTEX play a great role in the oil produces wastewater, which also contribute the significance of your study. Then, in the results and discussion section, authors should pay more attention on the BTEX or other unique organic pollutants degraders in each phase.

-Another serious concern is about the sampling time. Authors didn’t mention the sample time in the manuscript, which makes reviewer wondering the operational condition of WWTPs. As we know, the influent and operational condition may change during the WWTP operation. Whether these sample were collected at the steady phase? In addition, one sample would represent the overall picture of microbial communities in each treatment phase?

-The limitation of methodologies might shade some information. For instance, as shown in the Table 1 and Figure S1, the limited reads (<12000 reads) may be due to the 454 pyrosequencing. And the relative abundance of the microbial communities and function might be not accurate. Compared to the predict function, metagenome might be a better option for this kind of study. For instance, the anaerobic process such as sulfate and nitrite reduction were mainly occurred in the anaerobic phase (ABR) instead of aerobic phase (APR). The Fig3 showed even very higher relative abundance in the APRs, which will be misleading the audience.

Specific concerns:

Page 3, line 10-11: related citation should be provided

Page 4, line 8: What do you mean “activated microbes”?

Page 4-5, line 12- 26: FAPROTAX is a bioinformatic tool for the microbial function prediction. It’s not that much related to the research gap that addressed in your study. Therefore, this section could be further simplified. Instead, more attention should be paid to the unique habitat of “offshore oil produced wastewater treatment plants”.

Page 5, line 23: Plz further indicate the sampling time

Page 6, line 24: QIIME. What database you used to identify the OTUs?

Page 7, section 3.1: one problem is that you didn’t have parallel samples

Page 9, line 2-8: This information should be placed to Material and Method section

Page 9, line 19-22: several identified OTUs might not fully support your idea. Source tracker (Knights et al., 2011) might be a good tool to verify your statement as also shown in Highlight #1.

Knights, D., Kuczynski, J., Charlson, E.S., Zaneveld, J., Mozer, M.C., Collman, R.G., Bushman, F.D., Knight, R., Kelley, S.T., 2011. Bayesian community-wide culture-independent microbial source tracking. Nat. Methods 8, 761–763.

Page 9, line 25: WWTS? waste water treatment system? Better to rephrase this sentence, redundant words as “as microorganism in WWTPs”

Page 10, line 13-16: the statement needs evidence. Here, authors should describe what kind of bacteria was originated from oil reservoir and marine environment.

Page 12, line 11-17: As mentioned, authors should pay more attention on the degraders treating unique pollutants in the oil produced wastewater, instead of this not relevant information which cannot supported by current results.

Page 13, line 15: OTU instead of OUT, also check throughout the manuscript

Page 14, line 16: What is DO concentration in ABR?

Fig.2 It’s not a good idea to show the microbial communities by using pie chart. Hard to differentiae the color for each phylum. And what is the meaning of number after phylum? such as Bacteroidetes (5)

6. PLOS authors have the option to publish the peer review history of their article (what does this mean?). If published, this will include your full peer review and any attached files.

Reviewer #1: No

Reviewer #2: No

Reviewer #3: No

---

## [Author Response · Author response to Decision Letter 0]

18 Jun 2021

Reviewer 1:

Recommendation: This manuscript elucidated the microbial composition and the potential microbial functions in a full-scale well-worked offshore oil produced wastewater treatment plant using 16S rDNA amplicon sequencing. In general, the finding of this work is interesting. However, the following comments should be addressed.

(1) The level of English should be improved and the data format should be unified throughout the manuscript. 

Response:

We have polished our manuscript through the ShineWrite.com and have checked the words, sentences and data throughout the manuscript.

(2) In “Highlight” sections, their content should be rewritten and simplified for better description the manuscript. 

Response: 

Pg 2, Line 17. We have rewritten the “Highlight” according to the reviewer’s suggestion. 

(3) If the abbreviation appears for the first time in the manuscript, please indicate its specific name. (like VFA, WWTS)

Response:

Pg 6, Line 121. “VFAs” appears for the first time and is indicated.

Pg 11, Line 216. WWTS appears for the first time and is indicated.

We checked other abbreviations and indicated their special names in the manuscripit. 

(4) “NH3-N concentration in the water phase of ashore water-oil sample was 125.58 mg/L, after the anaerobic treatment the NH3-N concentration decreased to 56.72 mg/L”. Please explain how NH3-N was removed in ABR.

Response: 

Based on the data of microbial community and functional annotation in the ABR (Figure 3 and 4), we could not find typical microbes and functions of ammonia oxidation. Therefore, we could not give a concrete microbial reason on the decrease of NH3-N concentration in the ABR. As we discussed in Line 176-186, we found the inconsistency and only could give a general reason of many relative factors that varied in biological wastewater treatment processes. 

(5) Concentration of sulfate was increased in ABR, why? How COD was removed in ABR? Why was not methane produced in ABR?

Response: 

Pg 17, from line 359-387, we discussed the sulfur cycle and typical microbes in the ABR, sulfur cycle involving sulfate reduction to sulfide (SO42−→S2−) and subsequent sulfide oxidation to elemental sulfur (S2−→S0). Here, there wasn’t obvious microbial evidences for sulfate production (typical SOBs for sulfate). Uncultured Campylobacterales were dominant, however studies for their metabolism were few as the uncultured features. Oxygen introduction might be for chemical oxidation of sulfide，while the data we had was not adequate. Therefore, more studies are needed. Here we described/reported the microbial characteristics in the plant to provide a basic researching results for the following studies. 

Pg 18, 380-387, we referred to COD removal in the ABR as sulfur cycle as the study by Yun et al., 2019.

Fig 4. We did not detected methane and Methanogen in the study, but we mentioned the potential metabolism of methane production in Fig 4. Studies on archaea is our next focus point. 

(6) What was the physical processes?

Response:

Pg 101-113, Figure 1. We described and illustrated the each treatment process, before the wastewater coming to biotreatment, it flowed from the three separators, the horizontal oil eliminator, the vertical walnut shell filter and the settlement pond (SP). These are physical processes for oil produced wastewater. 

(7) The salt concentration in the oil production wastewater is high. In the process of biological treatment of wastewater, how much influence does it have on the treatment efficiency of microorganisms?

Response: 

Pg11, line 209-239. The salt concentration in the oil-produced wastewater is high as the seawater-flooding oil recovery and the produce of formation water. However, the influence of high salinity on microbes was low. As we found that a majority of the detected microbes in the plant were originated from oil reservoirs or marine-associated environments. Selective pressure imposed by environmental impactors shaped the microbial communities the plant.

(8) In three aerobic reactors, the diversity and functional similarities of microorganisms in three different aerobic treatments need to be discussed in more detail.

Response:

From Pg 20, Line 411. It is better to discuss the results in the three aerobic reactors in more detail. However, given the length of the manuscript, the numbers of dominant OTUs and the complex of microbial function net work, we summarized that different microbes play similar functions in the three aerobic parallel reactors in a whole. The discussion part showed the predominant microbial characteristics in the three APRS. 

(9) The layout of Figure 3 May be improve and some signs overlap so the signs are not clear.

Response: The relative abundances of functions in some samples show close, some signs in the Figure 3 overlapped. we have used hollow/soil/dash sphere/ triangle and colors to identified different samples and have tried many combination schemes. The present layout might be the most optimal show for the data with a little overlap. We will provide a higher resolution image for publication. 

(10) Please check “A majority of sequences detected by highthrough sequencing methods are identified with bacteria below to order lever that can not be affiliated with functional groups as many functions are only conserved at species or genus 

level in FAPROTAX database. I”.

Response:

Pg 4 line 79-83. Despite FAPPOTAX apparent merits of constructing prokaryotic tax to metabolic or other ecologically relevant functions, it has limitations. The statement at line 84-88 was cited from the reference (Louca et al., 2016). We revised the sentence and added the reference. 

Special thanks to you for your good comments 

Reviewer 2:

Recommendation: In this study authors describe the biological treatment of oil-produced wastewater. To improve the treatment process and meet stringent effluent discharge limits, it is important to identify the microbial members found in these systems and understand their metabolic potential. The topic is interesting, the experiments are comprehensive, the findings are useful, for adding in-depth understanding to biological wastewater treatment. To make it better, I have some suggestions/questions as below:

(1) Pg 3, ‘…remains curtail to modern civilization..” 

Sentence unclear

Response:

Pg 3 Line 40, in the sentence, we revised “crutail” to “important”.

(2) Pg 5, In this work, we elucidated the structures and potential functions of bacteria in a full-scale offshore oil produced wastewater treatment plant based on highthrough 16S rDNA sequences.

Do the authors mean 16S ribosomal RNA (rRNA) or recombinant DNA (rDNA)?

Response:

Pg 4, Line 72, here we mean16S ribosomal RNA (rRNA) gene. 

(3) Pg 6, correct OIIME-1.6.0 Pipelines to QIIME-1.6.0…

Response: 

Pg 7, Line 142, We have corrected “OIIME-1.6.0 Pipelines “ to “QIIME-1.6.0”.

(4) Pg 8, ‘NH3-N concentration in the water phase of ashore water-oil sample was 125.58 mg/L, after the anaerobic treatment the NH3-N concentration decreased to 56.72 mg/L, and after aerobic treatment the NH3-N concentration in the effluent below 1 mg/L (Table 1).’

Ammonia usually has an inhibitory effect on anaerobic digestion but given that 334 mg/L COD reduction was observed, it might seem that the ammonia was either being stripped in gas phase or some microorganisms present in the digester could utilize ammonia since in section 3.4 there is mention of oxygen intrusion? 

Response:

Pg 9 Line 176-178, we also found the the COD and NH3-N removal rates in our anaerobic and aerobic reactors were inconsistent with those in previous studies. However, based on the data of microbial community and functional annotation in the ABR (Figure 3 and 4), we could not find typical microbes and functions of anaerobic ammonia oxidation. Paradoxically, oxygen intrusion was mentioned as the appearance aerobic microbes. According to the data of detected microbes, we could not give a concrete microbial reason on the decrease of NH3-N concentration in the ABR and only could give a general reason of many relative factors that varied in biological wastewater treatment processes. More studied of uncultured microbes were needed. 

(5) Pg 8, Section 3.2, It is important to statistically analyse changes in abundance of microbial community members (Fig 2) and metabolic functions (Fig 3) in different treatments and include it in discussions of sections 3.2 – 3.5.

Response:

We format the results and discussion in one section in our manuscript to describe the microbial characteristics of our studied samples. As the numbers of detected dominant microbes and functions were abundant, and considering the length limitation of the manuscript, we presented the results where they were used to each subtitle. Therefore, Fig 2 was used dispersedly at Pg 10, Lin2 203, Pg 14, Line 300-304, Pg 15 Line 335, Pg 20, 415-420, 427-429. 

(6) Pg 9, ‘In addition, as seen in Fig. S1, only the rarefaction curve of AOW and SP tended to approach the saturation plateau. The rest of rarefaction curves show little sign of plateau, indicating that the biological treatment processes contained a somewhat broader phylogenetic diversity...’

α-, β diversity measures will be more conclusive than rarefraction curves. Rarefraction curves not plateauing may also refer to insufficient depth of sequencing. 

Response:

Pg 10, Line 211, It is true that rarefraction curves can show the diversity measures and the depth of sequencing. In the original version of our manuscript, we used the rarefraction curves to indicated that microbial diversity in biotreatment processes were higher than that in physical processes without considering the depth of sequencing. In order to avoid the dispute, we removed the results of rarefraction curves (Fig S1）. 

(7) Pg 12, ‘…which could provide microbial data to the downstream treatment system, especially to those systems using biotreatments.’

Sentence unclear

Response: 

Pg 14, Line 285, removed the sentence. 

(8) Pg 16, However, the aerobes and anaerobes co-existing ecology systems were reported popular in natural environments and could significantly enhance pollutants removal... 

Pg 17, …as organic matters from wastewater are initially adsorbed to the activated sludge and are degraded by microbes inhabited in the inner of the sludge (Mario, et al., 2019).

Above explanations can be improved by discussing factors that lead to ecological gradients in sludge and how this allows microorganisms to co-exist in these conditions. 

Response: 

Yes, it is true to make a good discussion point about relations between ecological gradients and microbial co-exist. However, our present version of the manuscript is very long, we have to soften some point. 

(9) Pg 20 - Hyphomicrobiaceae-affiliated microbes inhabited in wastewater system were commonly detected to posse nitrogen removal enzymes and were closely related to nitrogen removal (Jia et al., 2019; Tomasek et al., 2017).

Authors may also discuss that Hyphomicrobium-related species are also commonly found in denitrifying wastewater treatment systems as identified in:

a) Lu, H., Chandran, K. and Stensel, D., 2014. Microbial ecology of denitrification in biological wastewater treatment. Water research, 64, pp.237-254.

Response: 

Pg 223 Line 478-479, we added the reference. 

(10) Pg 20 – “However, as the microbial communities of the studied reactors were so complex and consisted of a large percentage of uncultured candidate divisions, it was not easy to draw the potential metabolize network in each reactor. More studies associated strains isolation and function genes detection are therefore important to follow up.”

Authors can mention that since many microorganisms could not be identified, it should be considered in future work on this topic. Further, strain isolation has several redundancies but new techniques such as long-read sequencing methods are being increasingly applied to identify microbial community members in wastewater treatment plants as shown in:

a) Singleton, C.M., Petriglieri, F., Kristensen, J.M., Kirkegaard, R.H., Michaelsen, T.Y., Andersen, M.H., Kondrotaite, Z., Karst, S.M., Dueholm, M.S., Nielsen, P.H. and Albertsen, M., 2021. Connecting structure to function with the recovery of over 1000 high-quality metagenome-assembled genomes from activated sludge using long-read sequencing. Nature Communications, 12(1), pp.1-13. 

b) Arumugam, K., Bessarab, I., Haryono, M.A.S. et al. 2021. Recovery of complete genomes and non-chromosomal replicons from activated sludge enrichment microbial communities with long read metagenome sequencing. npj Biofilms Microbiomes 7, Nature Publishing Group, https://doi.org/10.1038/s41522-021-00196-6

Response:

Pg 24, Lin 492-494, following the comments, we revised the paragraph and added the references.

(11) Pg 20 – Conclusion: In this study, we have characterized the composition and interaction of the bacteria inhabiting in the full-scale…

>In my understanding, no interactions have been mentioned in the manuscript, rather the functional potential is discussed.

Response:

Pg 23, Lin 497, we revised the sentence as “In this study, we characterized the composition and potential functions of ....”

(11) Table 1. Correct to AWO

Response:

We checked all ashore oil-water (AOW)

(12) Fig 1. Can mention in footnote that 2 samples, ABRS and ABRW, were collected from ABR.

Response:

Fig 1. we revised “AWO” to “AOW” and mentioned ABRS (Sludge sample) and ABRW (Water sample)

(13) Fig 2. Different colours to be used for alphaproteobacteria and unidentified taxa. 

Response: we used highcharts with dashes and colors 

References – Please check the consistency of all references, including author names, journal names and titles.

Response: we checked all the references. 

We tried our best to improve the manuscript and made some changes in the revised manuscript. These changes will not influence the content and framework of the paper.

We appreciate for Editors and Reviewers’ warm work earnestly, and hope that the correction will meet the approval. 

Once again, thank you very much for your comments and suggestions.

---

## [Decision Letter · Decision Letter 1]

26 Jul 2021

Elucidate microbial characteristics in a full-scale treatment plant for offshore oil produced wastewater

PONE-D-21-08773R1

Dear Dr. Zhang,

We’re pleased to inform you that your manuscript has been judged scientifically suitable for publication and will be formally accepted for publication once it meets all outstanding technical requirements.

Kind regards,

Guanglei Qiu

Academic Editor

PLOS ONE

Additional Editor Comments (optional):

Reviewers' comments:

Reviewer's Responses to Questions

**Comments to the Author**

1. If the authors have adequately addressed your comments raised in a previous round of review and you feel that this manuscript is now acceptable for publication, you may indicate that here to bypass the “Comments to the Author” section, enter your conflict of interest statement in the “Confidential to Editor” section, and submit your "Accept" recommendation.

Reviewer #1: All comments have been addressed

Reviewer #2: All comments have been addressed

2. Is the manuscript technically sound, and do the data support the conclusions?

Reviewer #1: Yes

Reviewer #2: Yes

3. Has the statistical analysis been performed appropriately and rigorously? 

Reviewer #1: Yes

Reviewer #2: Yes

4. Have the authors made all data underlying the findings in their manuscript fully available?

Reviewer #1: Yes

Reviewer #2: Yes

5. Is the manuscript presented in an intelligible fashion and written in standard English?

Reviewer #1: Yes

Reviewer #2: Yes

6. Review Comments to the Author

Reviewer #1: The authors have tried their best to solve all the problems. the manuscript is presented in an intelligible fashion and written in standard English.The authors have made all data underlying the findings in their manuscript fully available.

Reviewer #2: The authors have responded satisfactorily to the suggestions and have improved the manuscript. The MS now reads well and presents novel insights on the treatment of offshore oil-produced wastewater.

7. PLOS authors have the option to publish the peer review history of their article (what does this mean?). If published, this will include your full peer review and any attached files.

Reviewer #1: No

Reviewer #2: No

---

## [Editor Report · Acceptance letter]

5 Aug 2021

PONE-D-21-08773R1 

Elucidate microbial characteristics in a full-scale treatment plant for offshore oil produced wastewater 

Dear Dr. Zhang:

I'm pleased to inform you that your manuscript has been deemed suitable for publication in PLOS ONE. Congratulations! Your manuscript is now with our production department. 

Kind regards, 

on behalf of

Dr. Guanglei Qiu 

Academic Editor

PLOS ONE